# Overexpression of *HbGRF4* or *HbGRF4-HbGIF1* Chimera Improves the Efficiency of Somatic Embryogenesis in *Hevea brasiliensis*

**DOI:** 10.3390/ijms25052921

**Published:** 2024-03-02

**Authors:** Xiaomei Luo, Yi Zhang, Miaomiao Zhou, Kaiye Liu, Shengmin Zhang, De Ye, Chaorong Tang, Jie Cao

**Affiliations:** 1School of Breeding and Multiplication (Sanya Institute of Breeding and Multiplication), Hainan University, Sanya 572025, China; luoxiaomei@hainanu.edu.cn (X.L.); 182714@hainanu.edu.cn (Y.Z.); zhoumiaomiao@hainanu.edu.cn (M.Z.); kaiyeliu@hainanu.edu.cn (K.L.); zsm@hainanu.edu.cn (S.Z.); yede@cau.edu.cn (D.Y.); 2School of Tropical Agriculture and Forestry, Hainan University, Sanya 572025, China; 3National Key Laboratory for Biological Breeding of Tropical Crops, Hainan University, Haikou 570228, China; 4Natural Rubber Cooperative Innovation Center of Hainan Province and Ministry of Education of PRC, Hainan University, Haikou 570228, China

**Keywords:** somatic embryo production, rubber tree, *HbGRF4*, *HbGRF4-HbGIF1*

## Abstract

Transgenic technology is a crucial tool for gene functional analysis and targeted genetic modification in the para rubber tree (*Hevea brasiliensis*). However, low efficiency of plant regeneration via somatic embryogenesis remains a bottleneck of successful genetic transformation in *H. brasiliensis*. Enhancing expression of *GROWTH-REGULATING FACTOR 4* (*GRF4*)-*GRF-INTERACTING FACTOR 1* (*GIF1*) has been reported to significantly improve shoot and embryo regeneration in multiple crops. Here, we identified endogenous *HbGRF4* and *HbGIF1* from the rubber clone Reyan7-33-97, the expressions of which dramatically increased along with somatic embryo (SE) production. Intriguingly, overexpression of *HbGRF4* or *HbGRF4-HbGIF1* markedly enhanced the efficiency of embryogenesis in two *H. brasiliensis* callus lines with contrasting rates of SE production. Transcriptional profiling revealed that the genes involved in jasmonic acid response were up-regulated, whereas those in ethylene biosynthesis and response as well as the *S*-adenosylmethionine-dependent methyltransferase activity were down-regulated in *HbGRF4*- and *HbGRF4-HbGIF1*-overexpressing *H. brasiliensis* embryos. These findings open up a new avenue for improving SE production in rubber tree, and help to unravel the underlying mechanisms of HbGRF4-enhanced somatic embryogenesis.

## 1. Introduction

Natural rubber (NR, *cis*-1,4-polyisoprene) is a high molecular weight biopolymer consisting of a vast number of isoprene (C_5_H_8_) units linked one by one all in a 1,4-*cis*-configuration [1]. NR has multiple elite physical and mechanical properties, including high elasticity, resistance to abrasion and impact, efficient heat dispersion, and malleability at cold temperatures, which make it irreplaceable by synthetic alternatives in many important applications, such as aircraft tires [2]. Due to its high yield and easy harvesting, the rubber tree (*Hevea brasiliensis*) has become the sole commercial NR source despite the presence of over 2500 NR-producing plant species [3]. The global NR demand is expected to continually increase in the future [4]. In 2022, according to the report of Association of Natural Rubber Producing Countries (ANRPC), global NR production reached 14.343 megatons (mt) with a 4.03% increase over the previous year, while the consumption was 14.805 mt, 3.22% higher than the production. Rubber tree is a typical tropical species, which makes it difficult to increase rubber production by expanding growing areas to non-tropical regions [5]. Furthermore, fungal diseases constantly afflict the rubber production, especially the devastating South American Leaf Blight (SALB) that seriously prevents large-scale rubber production in South America and Latin America [6,7,8]. Therefore, breeding elite rubber clones with high yield and disease resistance is a reliable strategy to assure global NR security. However, the highly heterozygous nature, long juvenile phase, and prolonged period required for evaluation of mature traits are strong limitations confronted in conventional rubber breeding [9]. Transgenic technology has opened a gate to speeding-up the process of rubber breeding.

Successful rubber tree transformation was first reported by Arokiaraj et al. [10] using the particle gun method, and thereafter various studies have been reported on the effect of explants, transformation methods, plasmid usage, and cultural conditions on transgenic efficiency in rubber tree [11,12,13,14,15]. A reliable stepwise method of *Agrobacterium*-mediated genetic transformation was established in *H. brasiliensis*, including the steps of callus inoculation with *Agrobacterium*, cocultivation, selection of transformed callus, somatic embryogenesis, and rooting [16]. Among the steps of this method, somatic embryogenesis remains a major obstacle to successful transgenesis. Currently, there are two main methods of somatic embryogenesis in *H. brasiliensis*, i.e., somatic embryo (SE) regenerated from primary callus and from friable callus, both types of calli commonly deriving from internal integument of immature seeds or anthers [17,18]. Plantlets produced from primary callus are usually of good growth and high rubber production [19], but the efficiencies of callus multiplication and SE production are low [20]. By contract, the friable callus has higher embryogenic potential, and it can be maintained after numerous subcultures and long-term cryopreservation [21,22,23]. Moreover, friable callus lines can be rapidly established by using the fragments of embryos derived from primary callus [24]. Unfortunately, SE can only be regenerated from friable calli in just a few of rubber clones, and the efficiencies of embryo and plant regeneration are variable [25]. Even worse, friable callus lines derived from the same rubber clone vary markedly in SE production rates [19]. Therefore, new methods to improve somatic embryogenesis from rubber callus is important for efficient transgenesis in rubber tree.

Substantial improvements in somatic embryogenesis and de novo organogenesis have been achieved in a number of plant species by overexpressing several plant developmental regulators, including WUSCHEL (WUS), BABY BOOM (BBM), and GROWTH-REGULATING FACTOR-GRF-INTERACTING FACTOR (GRF-GIF) [26,27,28,29]. Overexpression of *WUS* and *BBM* promotes the regeneration of various transformation-recalcitrant plant genotypes, but constitutive expression of these two genes induced pleiotropic negative phenotypes, like sterility and severe dwarfism. For practical purpose, these expression cassettes have to be excised from engineered plants [30]. GRF transcription factors contain conserved WRC and QLQ domains in the N-terminal region, and GIF proteins are a class of transcriptional co-activators harboring conserved SNH domain; GRFs interact with GIFs through the interaction of QLQ and SNH domains to form a functional transcriptional complex (GRF-GIF) [31,32,33,34]. Overexpression of *Triticum aestivum GRF4-GIF1* (*TaGRF4*-*TaGIF1*) or its orthologs substantially increases the efficiencies of embryo and shoot regeneration in dicot and monocot plants, including wheat, rice, *Citrus*, grape, watermelon, and lettuce, and dramatically expand the scope of genotypes suitable for genetic transformation [35,36,37,38]. Meanwhile, the *GRF4-GIF1* transgenic plants show no obvious developmental defects. Ectopic expression of a single *GRF* gene, the *Arabidopsis GRF5* (*AtGRF5*) also positively boosted shoot regeneration in several plant species [39,40].

In the present study, we identified the *GRF4* and *GIF1* homologues in *H. brasiliensis* and investigated their efficacy in boosting somatic embryogenesis of *Hevea* callus. We demonstrated that either a *GRF4* homologue, *HbGRF4*, or its chimeric combination with a *GIF1* homologue (*HbGIF1*), *HbGRF4-HbGIF1*, can be effective boosters for somatic embryogenesis from *Hevea* callus. In addition, we observed differentially expressed genes associated with jasmonic acid response, ethylene biosynthesis and response, and *S*-adenosylmethionine-dependent methyltransferase activity in *HbGRF4*-overexpressing *Hevea* embryos compared to the control. Our results provide a potential avenue for improving the efficiency of genetic transformation and benefit the understanding of HbGRF4-induced somatic embryogenesis in *H. brasiliensis*.

## 2. Results

### 2.1. Identification of the GRF4 and GIF1 Homologues in H. brasiliensis

We identified a total of 16 *HbGRF* and 5 *HbGIF* genes from the rubber clone Reyan7-33-97, with all respective proteins containing characteristic domains, i.e., WRC and QLQ for HbGRFs (Figure 1a, Appendix A) and SNH for HbGIFs (Figure 1b, Appendix A). A phylogenetic tree was generated based on the protein sequences of GRF and GIF from *H. brasiliensis*, *Triticum aestivum,* and *Arabidopsis thaliana*. HbGRFs and HbGIFs were classified, respectively, into four (group I to IV) and two groups (group I and II) together with the counterparts of *T. aestivum* and *A. thaliana* (Figure 1). Of the HbGRFs and HbGIFs, HbGRF3 (EVM0011360.1) and HbGRF4 (EVM0012342.1) exhibited high similarity with TaGRF4, while HbGIF1 (EVM0008481.1) and HbGIF4 (EVM0037662.1) were grouped with AtGIF1 and TaGIF1 (Figure 1). Focus was then placed on the four *HbGRF* and *HbGIF* genes due to the reported efficacy of *TaGRF4* and *TaGIF1* for enhancing embryo and shoot regeneration in diverse plant species [35].

To investigate the potential roles of HbGRF3, HbGRF4, HbGIF1, and HbGIF4 during somatic embryogenesis, the expression levels of these genes were evaluated in calli (E0, E12), calli plus SEs (E33, E54), or SEs (E75) collected at different stages of somatic embryogenesis. To avoid the influence of callus adaptation and nutrient deprivation in the medium, samples were collected in the middle of each culture cycle (one culture cycle costed 21 days). Callus proliferation was cultured on subculture medium (SM) and then transferred to somatic embryo induction medium (SEIM) to facilitate embryo formation. No embryos were visible at the stages of E0 and E12, whereas the proembryos emerged at E33, the globular and pear-shaped embryos appeared at the E54, and the mature embryos were observed at E75 (Figure 2a). Due to the difficulty in completely separating embryos from callus at the stages of E33 and E54, a mixture of calli and embryos was harvested for analysis. The expression levels of *HbGRF3* and *HbGRF4* were low at the E0 stage, but increased dramatically from the onset of proembryos appearance, reaching a peak at E75 when mature embryos emerged (Figure 2b). Notably, *HbGRF4* exhibited significantly higher expression levels than *HbGRF3* at all stages, over 3-fold higher at the E75 stage. At the stages of E0 and E12, the expressions of *HbGIF4* were higher than those of *HbGIF1*; from the onset of proembryos, the expressions of *HbGIF1* and *HbGIF4* increased similarly to *HbGRF3* and *HbGRF4*, with *HbGIF1* showing a significantly higher level than that of *HbGIF4* at the stage of E75 (Figure 2c). These findings suggested that *HbGRF3*, *HbGRF4*, *HbGIF1,* and *HbGIF4* were involved in SE production, with *HbGRF4* and *HbGIF1* potentially playing more important roles in this process.

### 2.2. Overexpression of HbGRF4 or HbGRF4-HbGIF1 Promotes SE Production of Transgenic Callus Lines

Inner integuments of immature fruits from Reyan7-33-97 were sliced into thin sections to induce callus formation. The induced calli were then transferred to SM for generating fragile callus that shows yellowish, loose texture and granular fragility. A total of 12 fragile callus lines were selected, then transferred onto SEIM for somatic embryo (SE) regeneration (Appendix A). After 84 days of cultivation, several callus lines generated SEs, with the callus line C363 producing the highest number of embryos at an average SE production rate of 106.5 per gram of fresh calli (Appendix A). In addition, five fragile callus lines of C024, C148, C340, C363, and C366 were able to regenerate plantlets (Appendix A). 

To further investigate the roles of HbGRF4 and HbGIF1 in SE production, transgenic overexpression of *HbGRF4* and *HbGRF4-HbGIF1* was conducted in two fragile callus lines C073 and C363, with the lowest and highest rates of SE production, respectively. The callus lines were transformed with *A. tumefaciens* strain EHA105 harboring the overexpression plasmids of *HbGRF4* (p*35S*:*HbGRF4*), *HbGRF4*-*HbGIF1* (p*35S*:*HbGRF4-HbGIF1*), or the control vector p*35S*:*GFP* (Figure 3a). In all of these transformations, *Npt*II and *GFP* were employed as selection markers.

After co-culture and detoxification, the transformed callus lines were sub-cultured on selection medium, and only the paromomycin-resistant calli were viable. GFP activity was monitored by fluorescent protein excitation light source to select the fluorescent transgenic callus lines (Figure 3b). Then, 63 days after selective culture, we successfully obtained transgenic callus lines that were paromomycin-resistant and homogeneously GFP-fluorescent (Appendix A). From the callus line C073, two and eight independent transgenic lines were obtained, respectively, for the constructs of p*35S*:*HbGRF4* and p*35S*:*HbGRF4-HbGIF1*, and from the callus line C363, four and two transgenic lines were obtained, respectively, for the two constructs. A significantly higher expression of *HbGRF4* was detected by RT PCR in all p*35S*:*HbGRF4* and p*35S*:*HbGRF4-HbGIF1* transgenic callus lines compared to the p*35S*:*GFP* control line (Figure 3c,d).

In the low SE-regenerating callus line C073, overexpression of *HbGRF4* or *HbGRF4-HbGIF1* significantly increased somatic embryogenesis in all the independent transgenic callus lines, with the rates of SE production raised by 3.1–3.5 and 1.9–18.4 times, respectively, compared to the p*35S*:*GFP* control (Figure 4a,c). In the high SE-regenerating line C363, overexpression of *HbGRF4* or *HbGRF4-HbGIF1* resulted in a more effective effect on embryogenesis improvement, with the rates increased by 14.0–126.4-fold and 47.5–89.2-fold, respectively, for the two constructs (Figure 4b,d).

### 2.3. Transcriptome Analysis for the Mechanisms of HbGRF4-Promoting SE Production

To investigate the regulatory mechanisms of HbGRF4 on SE production in *H. brasiliensis*, we conducted RNA sequencing analysis on three transgenic C363 callus lines (363-p*35S*:*GFP*, 363-p*35S*:*HbGRF4*-3, and 363-p*35S*:*HbGRF4-HbGIF1*-1) at the E75 stage (Figure 5a). This stage was chosen because the highest endogenous expression of *HbGRF4* was observed during the process of somatic embryogenesis (Figure 2b). In the mature embryo of *HbGRF4*- or *HbGRF4-HbGIF1*-overexpressing line, the expression levels of *HbGRF4* were significantly elevated compared to the control (Figure 5b). Principal component analysis (PCA) revealed clear separation of the RNA-seq data among the three different transgenic lines, whereas the two data replications of the same transgenic line were near each other (Appendix A).

Compared to the 363-p*35S*:*GFP* line, the RNA-seq data identified 1361 differentially expressed genes (DEGs), 473 upregulated, and 888 down-regulated in the 363-p*35S*:*HbGRF4*-3 line; 1631 DEGs, 368 upregulated, and 1263 down-regulated in the 363-p*35S*:*HbGRF4-HbGIF1*-1 line. Among these DEGs, the two *HbGRF4*-overexpressing lines shared 148 upregulated and 645 down-regulated genes (Figure 5c, Appendix A). Gene Ontology (GO) analysis of the shared upregulated DEGs revealed enrichment in functional annotations associated with wounding response, jasmonic acid (JA) mediated signaling pathway, cell differentiation, and post-embryonic development. The shared down-regulated DEGs were enriched in abscisic acid transport, ethylene biosynthesis and response, and *S*-adenosylmenthionine-dependent methyltransferase activity (Appendix A). 

The expressional heatmap was presented for a number of up- and down-regulated DEGs of interest, including two JA responsive proteins: myelocytomatosis 2 (HbMYC2, EVM0021390.1) and jasmonate ZIM-domain 1 (HbJAZ1, EVM0009602.1), an ethylene-synthesizing enzyme (1-aminoacyl cyclopropane-1-carboxylic acid oxidase, HbACO4, EVM0007412.1), four ethylene response proteins (ERFs): HbERF1 (EVM0008798.1), HbERF5 (EVM0014526.1), HbERF098 (EVM0001731.1), and HbERF110 (EVM0018875.1), three *S*-adenosylmethionine-dependent methyltransferase (EVM0006996.1, EVM0021887.1, and EVM0026239.1), and an *O*-methyltransferase (EVM0028658.1) (Figure 5d,e). Among these genes, the expression patterns of eight were further validated by using RT PCR, the results of which were consistent with those shown in the RNA-seq analysis (Figure 6). These findings reinforced our hypothesis that HbGRF4 potentially plays a role in SE formation via the regulation of genes involved in JA response, ethylene biosynthesis and response, and *S*-adenosylmenthionine-dependent methyltransferase activity.

## 3. Discussion

*Agrobacterium*-mediated genetic transformation serves as a powerful tool in targeted modification of various crops, including the important industrial tropical crop species, *H. brasiliensis*. However, *H. brasiliensis* is recalcitrant to genetic transformation, and only a few clones (varieties) have so far been reported to be successfully transformed [41]. Low regeneration rates of somatic embryos and high genotype-dependence are the major limiting factors that hinder the efficient application of transgenic technique in rubber breeding. Carron et al. [18] examined the capability of embryogenesis in four industrial *H. brasiliensis* clones, including PB 260, PR 107, PB 235, and RRIM 600, under standard culturing conditions. Of the four clones, PB 260 exhibits the highest embryogenesis rate, while PR 107 and PB 235 display a medium rate, and RRIM 600 has lowest rate. The researchers on rubber genetic transformation are inspired by the findings of the past few years, through which substantial improvements on somatic embryogenesis were secured in diverse plant species by overexpressing some plant developmental regulators [35]. Notably, overexpression of *GRF4-GIF1* has been demonstrated to improve the regeneration efficiency of recalcitrant plant species, such as the hexaploid wheat varieties Hahn and Cadenza, as well as the triticale breeding line UC3190, and overexpression of *TaGRF4-TaGIF1* resulted in an increase in regeneration frequencies from 0% to 9–19% [34]. In this study, during the process of SE production in the rubber clone Reyan 7-33-97, we found SE production varied markedly among the different embryogenic callus lines generated from the inner integuments of immature fruits (Appendix A). Nevertheless, overexpression of the *HbGRF4* or *HbGRF4-HbGIF1* chimera significantly enhanced the rates of SE production in both the lowest (C073) and highest (C363) SE-regenerating callus lines examined (Figure 4c,d). These findings suggested that HbGRF4 and HbGRF4-HbGIF1 have the potential to expand the range of transformable genotypes in *H. brasiliensis*.

To investigate the mechanism underlying the enhancement of somatic embryo regeneration by HbGRF4, a transcriptomic analysis was performed comparing the *HbGRF4*- and *HbGRF4-HbGIF1*-overexpressing lines to the controls. The DEGs identified were enriched in JA mediated signaling pathway, ethylene synthesis and response, and *S*-adenosylmethionine-dependent methyltransferase activity (Figure 5d,e). The MYC2–JAZ1 feedback regulatory mechanism plays a curial role in JA response [42]. During the early stage of JA response, an increase in JA levels leads to the transient induction of MYC2, which subsequently up-regulates its own repressor, JAZ1. In turn, the up-regulation of JAZ1 suppresses MYC2 and triggers the late stage of JA response stage. The feedback loop helps to fine-tune the JA response and ensures appropriate timing and duration of the physiological responses associated with JA signaling. Intriguingly, in *Arabidopsis*, the repression of *MYC2* and the up-regulation of *JAZ1* lead to an increase in embryo production [43,44]. Here, we found by RNA-seq and RT PCR that the expressions of both *HbMYC2* and *HbJAZ1* were up-regulated in *HbGRF4*- and *HbGRF4*-*HbGIF1*-overexpressing lines (Figure 5d and Figure 6). Notably, *HbJAZ1* exhibited much higher expression levels than *HbMYC2*. These results suggested that HbGRF4 potentially modulates SE production by influencing the feedback regulatory loop of JA response involving HbMYC2 and HbJAZ1. 

In the rubber clones IRCA144 and PB260, addition of amino-oxyacetic acid or silver nitrate to the medium to inhibit ethylene synthesis or response has been found to improve SE formation [20]. Herein, the transcript abundance of *HbACO4*, an enzyme responsible for ethylene biosynthesis, was significantly decreased in both *HbGRF4*- and *HbGRF4-HbGIF1*-overexpressing lines (Figure 5e and Figure 6). Additionally, several ethylene-responsive transcription factors were also down-regulated in *HbGRF4*- and *HbGRF4-HbGIF1* overexpressing lines, including *HbERF1*, *HbERF5*, *HbERF098,* and *HbERF110*. *S*-adenosylmethionine-dependent methyltransferases utilize *S*-adenosylmethionine (SAM) as the methyl donor and transfer of methyl groups to various biomolecules, including DNA, RNA, proteins, and small-molecule secondary metabolites [45,46,47]. SAM-dependent methylation (SAMT) is involved in numerous important biological processes, including epigenetics and synthesis of a wide range of secondary metabolites, while the involvement of SAMT in somatic embryogenesis has not been reported. We revealed a significant decrease in the expressions of six genes enriched in *S*-adenosylmethionine-dependent methyltransferase activity in *HbGRF4*-overexpressing lines (Figure 5*e and*
Figure 6). These findings suggested that HbGRF4 might boost SE production by repressing ethylene biosynthesis and signaling, as well as *S*-adenosylmethionine-dependent methyltransferase activity.

In summary, our findings demonstrate that the overexpression of *HbGRF4* or *HbGRF4-HbGIF1* fusion significantly increases the efficiency of somatic embryo regeneration in different *Hevea* callus lines, and holds a promise in overcoming genotype-limited genetic transformation of rubber tree. 

## 4. Materials and Methods

### 4.1. Identification of HbGRF and HbGIF Genes and Phylogenetic Analysis

In order to identify HbGRF and HbGIF proteins, the amino acid sequences of TaGRF4 (TraesCS6A01G269600) and TaGIF1 (TraesCS4A02G250600) were employed as queries for a BLASP search in the local rubber tree genome database. The characteristic domains of WRC and QLQ for GRFs or SNH for GIFs were predicted for each retrieved protein using the Conserved Domain Database (CDD) (http://www.ncbi.nih.gov/cdd, accessed on 10 February 2024). The ID numbers and names of *HbGRF* and *HbGIF* genes are listed in Appendix A.

Full-length protein sequences of all GRF and GIF in wheat and *Arabidopsis* were downloaded from the NCBI database. Alignments of GRF and GIF protein sequences in *H. brasiliensis*, wheat, and *Arabidopsis* were performed with ClustalW algorithm of MEGA11 software (https://www.megasoftware.net/). A phylogenetic tree was constructed using the neighbor-joining method and bootstrapping with 1000 replicates. 

### 4.2. Selection of Embryogenic Fragile Callus Lines

Immature fruits of the *H. brasiliensis* clone Reyan7-33-97 were harvested from the field, and washed with distilled water in a laboratory. They were sterilized in 75% (*v*/*v*) ethanol for 2 min and 2% (*v*/*v*) sodium hypochlorite solution for 3.5 min, and then washed thoroughly in distilled water. Under sterile conditions, seeds were taken out from the immature fruits, which were thinly sliced and placed on callus induction medium (CIM) until calli generated from the inner integument tissues [48]. CIM was a modified maintenance culture medium (MM) supplemented with 4.65 μM Kinetin and eliminated AgNO_3_, abscisic acid (ABA), and benzylaminopurine (BAP). The calli were transferred to subculture medium (SM) with the same basic composition of MM medium but with 3 mM CaCl_2_; fragile callus with yellowish, loose texture, and granular fragility appeared after 42–126 days. Each fragile callus line was cut into small aggregates and cultivated on SM for further proliferation [48]. For somatic embryogenesis, an amount of 0.5–1 g of fragile callus from each line was placed on somatic embryo induction medium (SEIM) corresponding to a modified SM medium in which 3,4-D and BAP were reduced to 0.45 μM [48]. The average somatic embryo (SE) production rates (nb·g^−1^) were determined by measuring the total number of embryos per initial gram of fresh calli on SEIM for 64–84 days. All cultures were transferred to fresh medium every 21 days and carried out in dark conditions at 27 °C. SEs were subsequently cultured on the growth regulators-free medium (rooting medium) under a light intensity of 60 μmol m^−2^ s^−1^ and a 12 h light: 12 h dark photoperiod until rooting. Rooted embryos were transferred to glass tubes (200 mm × 50 mm) containing rooting medium (100 mL) to generate plants with fully expanded trilobed leaves [48].

### 4.3. Preparation of Plasmids and Agrobacterium Cell Suspension

The *HbGRF4* and *HbGIF1* genes were amplified by Polymerase Chain Reaction (PCR) from immature leaf cDNA of Reyan7-33-97. Then a *HbGRF4–HbGIF1* chimera was generated with a 4×*Ala* linker sequence by using overlapping PCR [36]. The terminal vector pCAMBIA2300 contains *green fluorescent protein* (*GFP*) and *neomycin phosphotransferase II* (*Npt*II) as selective genes driven by the *35S* promoter, respectively [48]. pCAMBIA2300 was linearized by restriction endonucleases *Sma* I and *Pst* I. *HbGRF4* and *HbGRF4–HbGIF1* were ligated to the linearized one by using ClonExpressII One Step Cloning Kit (Vazyme, Nanjing, China). The 4×*Ala* linker sequence and primers used in vector constructions are listed in Appendix A.

The recombinant plasmids and empty vector were introduced into *Agrobacterium tumefaciens* strain EHA105. The EHA105 strains carrying vectors were grown to OD_600_ = 0.6 in liquid Luria–Bertani medium (LB) with 50 mg L^−1^ kanamycin and 25 mg L^−1^ rifampicin, and then were diluted to OD_600_ = 0.1–0.2 with liquid subculture medium (LSM) as *Agrobacterium* cell suspensions for the following callus inoculation.

### 4.4. Inoculation, Cocultivation, Selection of Transgenic Callus Lines, and Somatic Embryo Production

The embryogenic calli were pre-cultured on calcium-free SM medium for 12 days [48]. Precultured calli were then immersed for 5–10 s in the *Agrobacterium* cell suspensions. After removing excess suspension, the calli were co-cultured in the dark at 20 °C for 3–5 days, and then transferred to the decontamination medium (DM), which was a modified SM medium supplemented with 450 mg L^−1^ timentin for inhibiting the growth of *Agrobacterium* cells [48]. After 42 days on DM, the new aggregates that had grown on the surface of calli were transferred to the fresh DM supplemented with 200 mg L^−1^ paromomycin, which was named as selection medium [48]. Paromomycin-resistant calli were assayed for GFP activity by the LUYOR-3260 fluorescent protein excitation light source (LUYOR, Shanghai, China) to identify pure fluorescent callus lines. Subsequently, an amount of 0.3–1.5 g pure GFP calli of each transgenic line was transferred to SEIM supplemented with 200 mg L^−1^ paromomycin for somatic embryogenesis. The average SE production rates were calculated as previously described. The initial callus weight, the number of embryos, and SE production rates of each transgenic callus lines are listed in Appendix A.

### 4.5. RNA Extraction and Quantitative Real-Time PCR (RT PCR) Analysis

Total RNA was extracted from the calli, embryos, and a mixture of calli and embryos according to the manufacturer’s instructions of Takara RNAiso Plus (Takara Biotechnology, Dalian, China). The RNA quality was examined by gel electrophoresis, and the concentration was quantified using Infinite^®^ 200 PRO (Tecan, Mannedorf, Switzerland). The cDNA was synthesized with a Vazyme HiScript III 1st Strand cDNA Synthesis Kit (+gDNA wiper). RT PCR was performed using the qTOWER^3^ G (Analytik Jena, Jena, Germany) in a 20 μL reaction containing 0.4 μL 10 μmol/L specific primers, 1 μL cDNA template, 8.2 μL ddH_2_O, and 10 μL Vazyme ChamQ Universal SYBR qPCR Master Mix (Vazyme, Nanjing, China). The RT PCR conditions were 95 °C for 30 s, followed by 40 repeated cycles of 95 °C for 10 s and 60 °C for 30 s. A melting curve was performed for each sample to confirm the specificity of the reactions. All RT PCR analyses were performed three times for each sample. The relative expression levels of each gene were normalized to the CT values of the *Hevea* reference gene *HbRh2b* (*HQ323243*) and calculated using the 2^−ΔCT^ method [49]. The primers used in the experiment are listed in Appendix A.

### 4.6. RNA Sequencing Analysis

Mature embryos with pure GFP signals were taken from transgenic line 363-p*35S*:*GFP*, 363-p*35S*:*HbGRF4*-3, and 363-p*35S*:*HbGRF4-HbGIF1*-1, respectively. The method of RNA extraction was similar to that previously described. The quality and concentration of RNA samples were examined by Agilent 5400 and NanoDrop. mRNA libraries were constructed by using the NEBNext Ultra RNA Library Prep Kit for Illumina (NEB, Ipswich, MA, USA, Catalog#:E7530L) following the manufacturer’s instructions. RNA sequencing was performed using the Illumina NovaSeqTM 6000 platform to obtain 150 bp paired end sequences. HISAT2-Stringtie pipeline was used to conduct transcriptomic analysis, and gene expression levels were normalized as fragments per kilobase of transcript per million mapped reads (FPKM).

Differentially expressed genes in 363-p*35S*:*HbGRF4*-3 and 363-p*35S*:*HbGRF4-HbGIF1*-1 compared to p*35S*:*GFP* were identified and screened using the DEseq2 package in R/Bioconductor with |log_2_foldchange| > 1 and padj < 0.05. Principal component analysis (PCA) and Gene Ontology (GO) term annotation were performed using OmicShare tools (https://www.omicshare.com/tools).

## Figures and Tables

**Figure 1 ijms-25-02921-f001:**
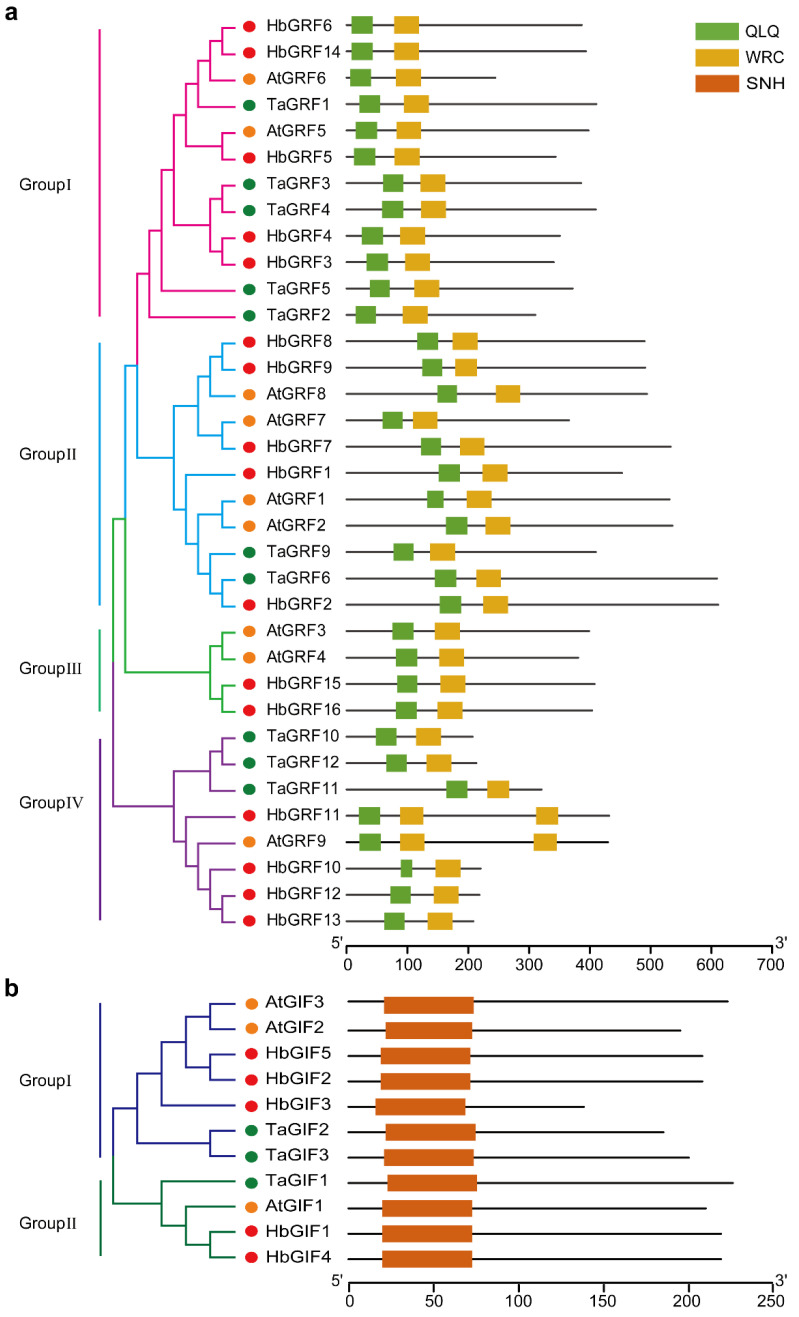
Phylogenetic relationships and conserved domain distributions of GRF and GIF proteins in *Hevea*, wheat, and *Arabidopsis*. (**a**) Phylogenetic tree and WRC and QLQ domain distributions of GRF in *Hevea*, wheat, and *Arabidopsis*. (**b**) Phylogenetic tree and SNH domain distribution of GIF in *Hevea*, wheat, and *Arabidopsis*.

**Figure 2 ijms-25-02921-f002:**
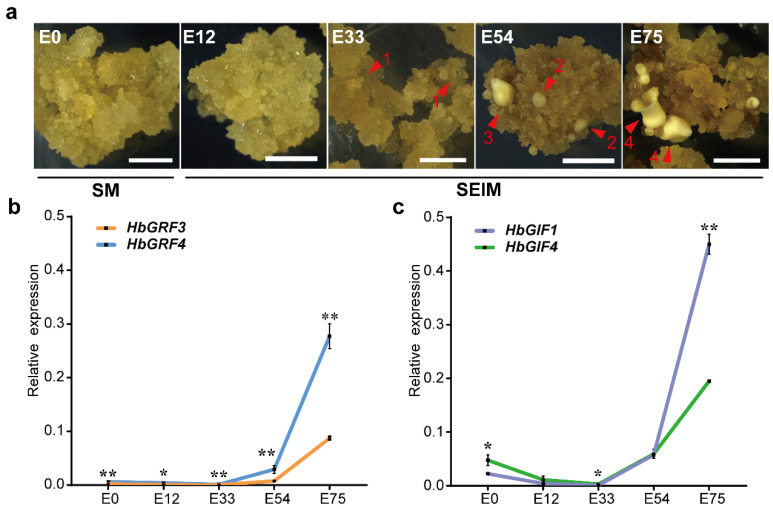
The expression levels of *HbGRF4* and *HbGIF1* were increased during somatic embryogenesis in *H. brasiliensis*. (**a**) Embryo regeneration phenotypes of *Hevea* callus. E0 is the stage before somatic embryo regeneration; E12, E33, E54, and E75 refer to the stages of somatic embryogenesis at 12, 33, 54, and 75 days; SM means subculture medium; SEIM means somatic embryo induction medium; arrow 1 indicates proembryo; arrow 2 indicates globular embryo; arrow 3 indicates the pear-shaped embryo; arrow 4 indicates mature embryo; bar = 2 mm. (**b**,**c**) The expression patterns of *HbGRF3*, *HbGRF4*, *HbGIF1*, and *HbGIF4* during somatic embryogenesis in *H. brasiliensis*. Mean ± SD, *n* = 3 technological repeats. Statistical significances were determined using Student’s *t*-test (* *p* < 0.05, ** *p* < 0.01, difference from *HbGRF3* and *HbGIF4*, respectively).

**Figure 3 ijms-25-02921-f003:**
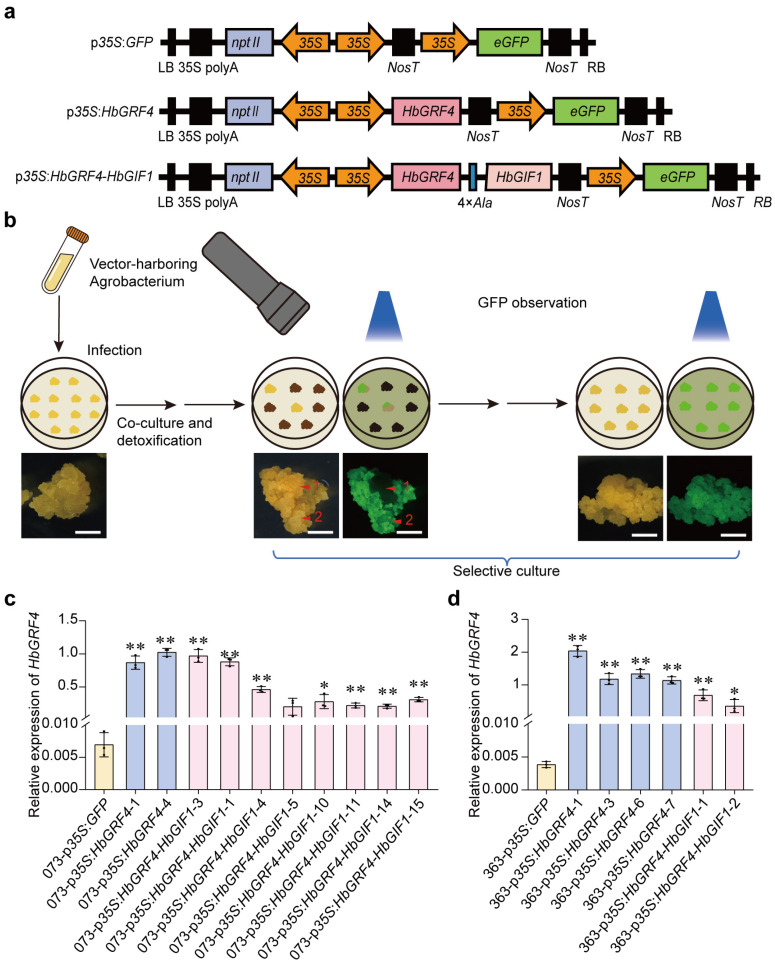
Identification of *HbGRF4*- and *HbGRF4*-*HbGIF*-overexpressing transgenic lines. (**a**) Schematic representation of *HbGRF4*- and *HbGRF4-HbGIF1*-overexpressing vectors; p*35S*:*GFP* was used as a control. (**b**) A sketch map of selecting paromomycin-resistant and pure GFP-fluorescent transgenic callus lines. Arrow 1 indicates non-fluorescent callus; arrow 2 indicates GFP-fluorescent callus; bar = 2 mm. (**c**,**d**) The relative expressions of *HbGRF4* in *HbGRF4*- and *HbGRF4-HbGIF1*-overexpressing callus lines and controls. Mean ± SD, *n* = 3 technological repeats. Statistical significances were determined using Student’s *t*-test (* *p* < 0.05, ** *p* < 0.01, difference from 073-p*35S*:*GFP* and 363-p*35S*:*GFP,* respectively).

**Figure 4 ijms-25-02921-f004:**
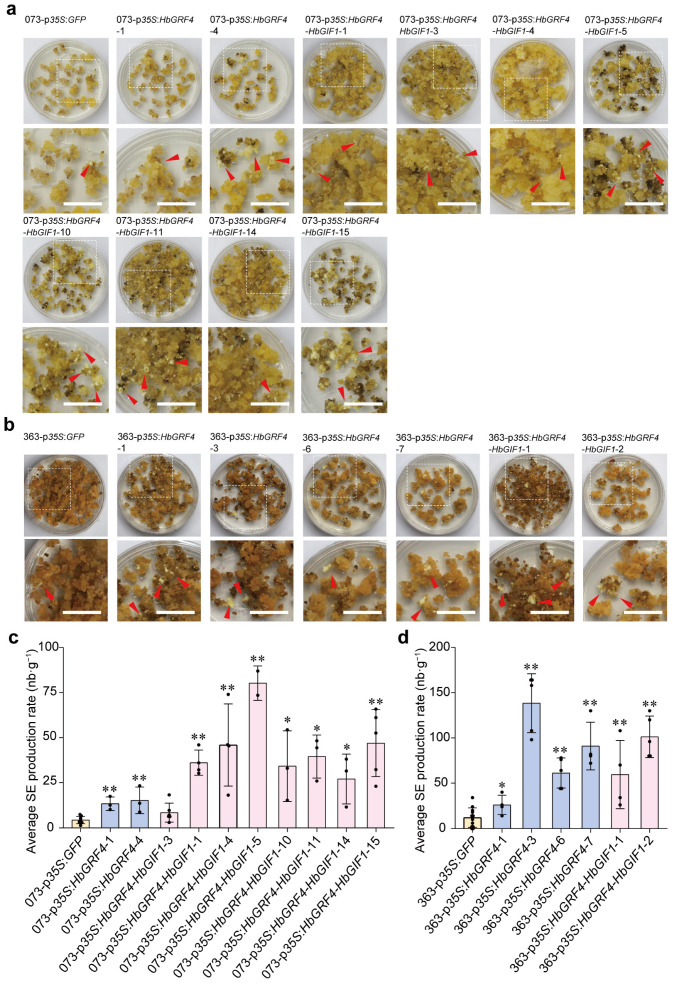
Overexpression of *HbGRF4* and *HbGRF4–HbGIF1* improves somatic embryogenesis in *H. brasiliensis*. (**a**,**b**) Embryo regeneration phenotypes of *HbGRF4*- and *HbGRF4*-*HbGIF1*-overexpressing lines and their controls. The area outlined by the white box in the top images are enlarged in the bottom images. Arrows indicate somatic embryos; bar = 2 cm. (**c**,**d**) The average SE production rates of *HbGRF4*- and *HbGRF4*-*HbGIF1*-overexpressing lines and their controls. Mean ± SD, *n* = 3–12 technological repeats. Results from individual experiment are indicated by black spots. The average SE production rate (nb·g^−1^) was calculated as the number of total embryos/gram of calli. Statistical significances were determined using Student’s *t*-test (* *p* < 0.05, ** *p* < 0.01, difference from 073-p*35S*:*GFP* and 363-p*35S*:*GFP*, respectively).

**Figure 5 ijms-25-02921-f005:**
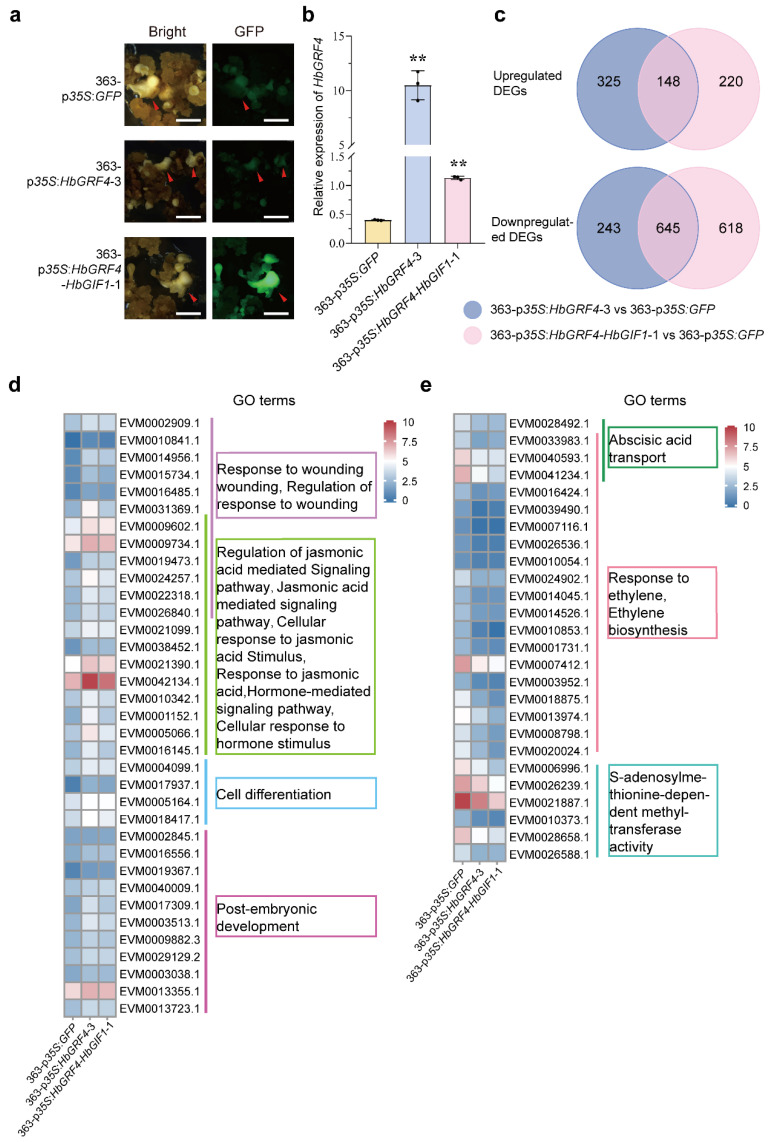
Differentially expressed genes associated with SE production in *HbGRF4*- and *HbGRF4*-*HbGIF1*-overexpressing lines compared to the control. (**a**) GFP signals of mature embryos regenerated from 363-p*35S*:*GFP*, 363-p*35S*:*HbGRF4*-3, and 363-p*35S*:*HbGRF4-HbGIF1*-1 callus lines. Red arrows indicate somatic embryos; bar = 2 cm. (**b**) Relative expression analysis of *HbGRF4* in (**a**). Mean ± SD, *n* = 3 technological repeats. Statistical significances were determined using Student’s *t*-test (** *p* < 0.01, difference from 363-p*35S*:*GFP*). (**c**) Venn diagram showing the number of DEGs in 363-p*35S*:*HbGRF4*-3 and 363-p*35S*:*HbGRF4-HbGIF1*-1 line compared to 363-p*35S*:*GFP*. |log2fold| > 1 and q < 0.05. (**d**,**e**) Heatmap and GO analysis of DEGs in 363-p*35S*:*HbGRF4*-3 and 363-p*35S*:*HbGRF4-HbGIF1*-1 line compared to 363-p*35S*:*GFP*.

**Figure 6 ijms-25-02921-f006:**
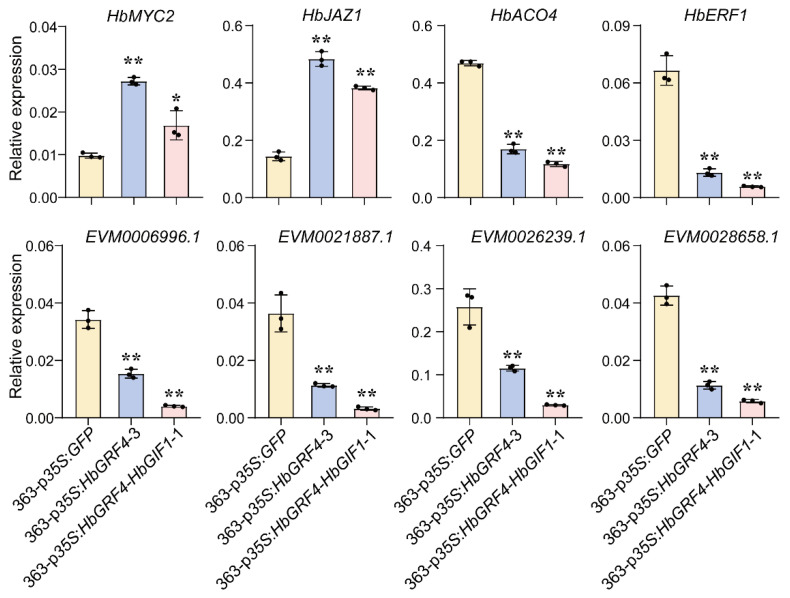
The relative expressions of eight DEGs involved in JA response, ethylene biosynthesis and response, and *S*-adenosylmenthionine-dependent methyltransferase activity. Mean ± SD, *n* = 3 technological repeats. Results from individual experiment are indicated by black spots. Statistical significances were determined using Student’s *t*-test (* *p* < 0.05, ** *p* < 0.01, difference from 363-p*35S*:*GFP*).

## Data Availability

The transcriptome sequencing data have been deposited in the Genome Sequence Archive in National Genomics Data Center (https://ngdc.cncb.ac.cn/, accessed on 10 February 2024) under accession number PRJCA022995.

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
