# Peer review of "Overexpression of HbGRF4 or HbGRF4-HbGIF1 Chimera Improves the Efficiency of Somatic Embryogenesis in Hevea brasiliensis"

_ijms, 2024, doi:10.3390/ijms25052921_

Round 1

Reviewer 1 Report

Comments and Suggestions for Authors

Comments on the Quality of English Language

Reviewer 2 Report

Comments and Suggestions for Authors

The manuscript submitted by Cao et al. introduces a study that aims to enhance the embryogenic cell formation of Hevea brasiliensis by overexpressing either HbGRF4 or HbGRF4-HbGIF1 chimera. Subsequently, the authors conducted transcriptomic analysis to investigate the impact of this mechanism on gene expression in Hevea brasiliensis, focusing specifically on data analysis and qPCR validation. Overall, this research contributes to an important area in transgenic biotechnology of Hevea brasiliensis. By leveraging known molecular mechanisms, the authors provide both practical evidence and theoretical exploration in transgenic research on Hevea brasiliensis.

Furthermore, I suggest that the authors conduct additional experiments involving the treatment of Hevea brasiliensis callus tissues with jasmonic acid and ethylene. This would help verify whether downstream genes, regulated by the mechanism identified in the transcriptomic analysis, are indeed involved in the action of these two hormone pathways.

Comments on the Quality of English Language

In my opinion, the overall flow of the authors' English writing is relatively smooth. However, there may be some individual word choices that could benefit from the assistance of native English-speaking authors to adjust.

Reviewer 3 Report

Comments and Suggestions for Authors

This article validates the reported regeneration gene GRF4-GIF1 in rubber trees, and the experimental data is solid. It has laid a certain foundation for the transformation of rubber trees and is a meaningful work. However, in the results section, I have a few small questions, as follows:

1: E0, E12, E33, E54, and E75 refer to the days 0, 12, 33, 54, and 75 after conversion? If so, an explanation should be provided in the text. If not, an explanation should also be provided. Also, why choose these time points.

2: I am not very familiar with rubber tree transformation receptor materials. I would like to ask why I chose Reyan7-33-97 as the material?

3: C024, C148, C340, C363, and C366 represent different lines of callus? Are all materials from Reyan7-33-97? Why not choose more rubber tree materials for callus treatment?
